# Kinetics Growth and Recovery of Valuable Nutrients from Selangor Peat Swamp and Pristine Forest Soils Using Different Extraction Methods as Potential Microalgae Growth Enhancers

**DOI:** 10.3390/molecules26030653

**Published:** 2021-01-27

**Authors:** Nor Suhaila Yaacob, Mohd Fadzli Ahmad, Nobuyuki Kawasaki, Maegala Nallapan Maniyam, Hasdianty Abdullah, Emi Fazlina Hashim, Fridelina Sjahrir, Wan Muhammad Ikram Wan Mohd Zamri, Kazuhiro Komatsu, Victor S. Kuwahara

**Affiliations:** 1Institute of Bio-IT Selangor, Universiti Selangor, Jalan Zirkon A7/A, Seksyen 7, Shah Alam 40000, Selangor, Malaysia; maegala@unisel.edu.my (M.N.M.); dianty@unisel.edu.my (H.A.); 2Centre for Foundation and General Studies, Universiti Selangor, Jalan Zirkon A7/A, Seksyen 7, Shah Alam 40000, Selangor, Malaysia; 3Department of Science & Biotechnology, Faculty of Engineering & Life Sciences, Universiti Selangor, Bestari Jaya 45600, Selangor, Malaysia; fadzli@unisel.edu.my (M.F.A.); hashim.emifazlina@nies.go.jp (E.F.H.); fridelina@unisel.edu.my (F.S.); ikramzamri1995@gmail.com (W.M.I.W.M.Z.); 4Dainippon Ink and Chemicals DIC Corporation, Central Research Laboratories, 631 Sakado, Sakura, Chiba 285-8668, Japan; nobuyuki-kawasaki@ma.dic.co.jp; 5Faculty of Education & Graduate School of Engineering, Soka University, 1-236 Tangi-Machi, Hachioji-Shi 192-8577, Japan; victor@soka.ac.jp; 6National Institute for Environmental Studies, 16-2 Onogawa, Tsukuba, Ibaraki 305-8506, Japan; kkomatsu@nies.go.jp

**Keywords:** biocatalysis, *Chlorella vulgaris* (TRG 4C), dissolved organic carbon, soil extraction, total dissolved nitrogen, total dissolved phosphorous, *Nannochloropsis oceanica* (TRG 3A)

## Abstract

Soil extracts are useful nutrients to enhance the growth of microalgae. Therefore, the present study attempts for the use of virgin soils from Peninsular Malaysia as growth enhancer. Soils collected from Raja Musa Forest Reserve (RMFR) and Ayer Hitam Forest Reserve (AHFR) were treated using different extraction methods. The total dissolved nitrogen (TDN), total dissolved phosphorus (TDP), and dissolved organic carbon (DOC) concentrations in the autoclave methods were relatively higher than natural extraction with up to 132.0 mg N/L, 10.7 mg P/L, and 2629 mg C/L, respectively for RMFR. The results of TDN, TDP, and DOC suggested that the best extraction methods are autoclaved at 121 °C twice with increasing 87%, 84%, and 95%, respectively. *Chlorella vulgaris* TRG 4C dominated the growth at 121 °C twice extraction method in the RMRF and AHRF samples, with increasing 54.3% and 14%, respectively. The specific growth rate (µ) of both microalgae were relatively higher, 0.23 d^−1^ in the Ayer Hitam Soil. This extract served well as a microalgal growth promoter, reducing the cost and the needs for synthetic medium. Mass production of microalgae as aquatic feed will be attempted eventually. The high recovery rate of nutrients has a huge potential to serve as a growth promoter for microalgae.

## 1. Introduction

Microalgae have many beneficial industrial applications especially in the formulation of cosmetics. Therefore, there is a need to uncover natural and cost-effective nutrient supplements that will enhance microalgae growth in large scale. Soil contains many valuable nutrients, and the present study embarks on the discovery of soil extracts from local habitats which could support the growth of microalgae. This is the first study to the best of our knowledge where reserved forest soils were used as nutrient sources for microalgal growth. Previous studies have established that synthetic nitrogen source was an important factor and indeed improved the growth of microalgae. The use of KNO_3_ particularly resulted in an increase of approximately 300% of biomass (g/L) compared to that of the control system, i.e., without the supplementing of nitrogen sources [1]. This study also reported that potassium and nitrogen played a vital role as nutrients to enhance microalgae growth. The information compelled and prompted the present study’s design to use soil extracts that contain a high concentration of those nutrients as an alternative source to support the growth of microalgae. Soil plays many important roles to support all living beings on earth. The thin layers of organic and inorganic materials on the surface of the earth provide a medium for plant growth and supports food production. Soil also contributes to the regulation of water, global climate, and atmospheric gases [2]. Soil serves as a habitat for many living organisms such as bacteria, fungi, and protozoans, and thousands of insects and worms. Meanwhile, water from rainfall and snowmelts got infiltrated into the ground which filters dust, chemicals, and other contaminants from underground water, rendering it one of the purest water sources. Besides, soil helps regulate atmospheric carbon dioxide (CO_2_) by acting as carbon storage [3]. To achieve an optimum final cell concentration, the culture medium formula is essential in microalgae cultivation. In addition, the culture medium components must fulfil the specific requirements for the build-up of marine microalgae cells and the production of metabolites by supplying an adequate energy supply for the metabolism reaction of marine microalgae. Some researchers have suggested that it would be useful for microalgae production to use microelements in soil extract. Moreover, several experiments have shown that soil extracts initiate rapid reproduction and growth in some microalgae [4].

Soil nutrients such as phosphorus, nitrogen, and trace metals are stored, transformed, and cycled in the soil. These nutrients have been extracted and used as soil extracts to culture many soil bacteria [5,6,7] and microalgae [8]. More compounds and molecules can be released to the solvent because the heat can provide high energy to break down compounds and molecules in soil which are released to the solvent. Autoclave method has been mainly used for the extraction medium for incubation of microalgae.

Nutrient released in the medium play a role in enhancing the microalgae productivity. Chemical conditions are the most important parameters affecting the growth of microalgae after the required cultivation regime. Microalgal species use nutrients from the medium and then recycle them into biomass, which is then further converted into energy and other raw chemical products [9,10]. The three key nutrients for microalgae growth are carbon, nitrogen, and phosphorus. The assimilation of these nutrients is highly influenced by the overall nutrient composition available in the medium of cultivation [11,12]. The organic matter of the soil consists of a number of elements, such as carbon (C), hydrogen (H), oxygen (O), magnesium (Mg), and small quantities of sulfur (S), nitrogen (N) and potassium (K), phosphorus (P) and calcium (Ca) [13]. Studied by [14] N applied with Rh seed inoculation at 100 percent of the recommended dose greatly increased crop yield, yield-attributing characteristics, seed quality, and main nutrient uptake. Although the use of higher rates of N adversely affected the performance of root nodulation and N fertilizer use, it maximized the economic returns in terms of net sales, cost profit ratio, and cost-benefit incremental ratio. Furthermore, the rationale behind the use of mixotrophic micro-algae for wastewater treatment lies in their ability to use organic and inorganic carbon, as well as inorganic carbon N and P in wastewater for their growing, resulting in reductions of substances concentration in water [12]. Ref. [15] reported that absence of nitrate (no nitrogen) leads to the least amount of biomass growth and production of microalgae. Subsequently, ref. [16] found that the addition of organic manures enhanced the growth and yield of maize crop. Ref. [17] proposed that exogenous application of nitric could be a practical and effective approach to mitigate the adverse effects of chilling stress and, subsequently, to ensure sustainable rice production.

Total carbon in the soil refers to the joint organic and inorganic carbon found in the soil layer [18]. Microalgae can be grown on inorganic carbon sources (dissociated form (HCO_3_^−^, CO_3_^2−^) and undissociated type (CO_2_, H_2_CO_3_) [19]. It is also capable of utilizing N from a number of inorganic substances (e.g., NH^4+^, NO_3_, and NO_2_) and organic sources (for example, amino acids, urea, purines, and nucleosides) [12,20,21]. Phosphorus is also an essential element for micro-algae that is involved in countless metabolic pathways as well as a structural component of phospholipids, nucleotides, and ATP. The co-limitation of N-P or the ratio of N:P tends to have a higher effect on the biochemical composition of algae than that of N or P alone [20]. Atomic carbon (C), N, P, and nutrient uptake in microalgal strains are also greatly affected by the nutrient level [22].

The aims of the present study are: (1) To evaluate the recovery of the nutrient components namely carbon, nitrogen, and phosphorus (CNP) from virgin soil extracts; and (2) to investigate the effect of the virgin soil extracts on targeted microalgae growth as alternative nutritional source. Hence, this study provides important findings on CNP content which is necessary to understand the growth enhancement of our targeted microalgae. In this study, two different soils collected from Raja Musa Forest Reserve (RMFR) and Ayer Hitam Forest Reserve (AHFR) in Malaysia were used to test the recovery of CNP in soil extracts using nine different extraction methods. The ability of different soil types to influence the recovery of CNP was also tested. Soil extracts from two different reserved forests, Raja Musa (RM) and Ayer Hitam (AH) were sampled to determine their natural growth-promoting effects on microalgae and also to evaluate the growth effects of soil extracts (SEs) on specific microalgae species. The recovered CNP concentrations and ratios could provide important information to understand the growth factors of targeted microorganisms and decrease the medium cost. High level of CNP highly supported the growth of microalgae and CNP in the undisturbed soil region is expected to rise because of the natural metabolism of flora and fauna.

## 2. Results

### 2.1. Total Dissolved Nitrogen (TDN), Total Dissolved Phosphorus (TDP), and Dissolved Organic Carbon (DOC) Concentrations in Soil Extracts with Natural Extractions

The TDN, TDP, and DOC concentrations in soil extract with different extraction methods are shown in Figures 3–5. The TDN concentrations in natural extraction are generally very low, ranging between 10.0 and 16.3 mg N/L and between 1.8 and 2.1 mg N/L for RMFR and AHFR, respectively (Figure 1). The slight increase of TDN observed from 1 h to 4 h for RMFR, but the value decreased between 4 h and 24 h. For AHFR, the values slightly decreased from 1 h to 24 h. Similar trends are also observed in TDP and DOC concentrations. The TDP concentrations in natural extraction ranged between 0.9 and 1.7 mg P/L and 0.15 and 0.19 mg P/L for RMFR and AHFR, respectively (Figure 2). The TDP concentrations decreased between 1 h and 4 h in both soil types, then almost no changes is observed between 4 h and 24 h. The DOC concentrations ranged between 106.4 and 144 mg C/L and 16.0 and 17.2 mg C/L for RMFR and AHFR, respectively (Figure 3). The slight increase of DOC is observed between 1 h and 24 h in both soils samples. All results indicated that the RMFR nutrient content is higher compared to AHFR in all analyses conducted. In the meantime, the DOC concentrations obtained using natural extraction methods showed no significant (*p* > 0.05) difference by statistical analysis ANOVA and *t*-test.

### 2.2. Total Dissolved Nitrogen (TDN), Total Dissolved Phosphorus (TDP), and Dissolved Organic Carbon (DOC) Concentrations in Soil Extracts with Autoclave Extractions

The TDN, TDP, and DOC concentrations of soil extracts using autoclave methods are much higher than those from natural extraction methods (Figure 1, Figure 2 and Figure 3). The TDN concentrations in the autoclave methods ranged between 45.2 and 132.0 mg N/L and 5.4 and 13.0 mg N/L for RMFR and AHFR, respectively (Figure 1). When compared between the 105 °C and 121 °C autoclave methods, the TDN concentrations are 70–120% higher in the 121 °C than those in the 105 °C for RMFR, and 25–80% higher for AHFR. The TDN concentrations are 30–40% higher in the twice-autoclaved than those in the once-autoclaved for the RMFR samples. Significant differences (*p* < 0.05) are noted using ANOVA and Turkey post-hoc between different autoclave methods used in this study.

Similar trend is also observed in the TDP concentrations in the autoclave extraction methods. The TDP concentrations ranged between 6.2 and 10.7 mg P/L and 0.34 and 0.72 mg P/L for RMFR and AHFR, respectively (Figure 2). The TDP concentrations are 10%–40% higher in 121 °C method than those in the 105 °C method for RMFR, and 30–70% higher for AHFR. The TDP concentrations are 15%–40% higher in the twice-autoclaved than those in once-autoclaved for both RMFR and AHFR. Significant differences (*p* < 0.05) using ANOVA and Turkey post-hoc of TDP concentrations are also observed between all extraction methods used.

The DOC concentrations ranging between 999 mg C/L and 2629 mg C/L and between 54.4 mg C/L and 170 mg C/L for RMFR and AHFR, are shown in Figure 3. The DOC concentrations are 20%–100% higher in the 121 °C method than those in the 105 °C method for RMFR, and 60%–90% higher for AHFR. Meanwhile, the DOC concentrations are observed to be 15%–25% higher in twice-autoclaved than those in once-autoclaved except 110% higher in 121 °C in the RMFR sample. Statistical analysis showed significant differences (*p* < 0.05) using ANOVA and Turkey post-hoc between different autoclave methods used in this study. Almost all results presented in this study demonstrated that the soil sample in the 120 °C twice autoclaved method showed the highest nutrient extraction with the RMFR sample as the best soils to be used for microalgae growth enhancement.

### 2.3. Comparison of Carbon, Nitrogen, and Phosphorus Recovery

Our results revealed that the recovery of nutrient from soil extracts obtained from RMFR is much higher than those from AHFR. Table 1 shows the ratios of CNP concentrations in RMFR and AHFR in nine different extraction methods. The recovery of TDN, TDP, and DOC is 4.7 to 13.6 times, and 6 to 19.7 times, and 6.7 to 18.3 times higher in RMFR than AHFR. The ratios of RMFR to AHFR in natural extraction ranged between 4.7 and 9.2, while those in autoclave methods ranged between 7 and 19.7. The ratios in the autoclave methods are higher than the natural extraction. The results indicated that soil twice-autoclaved at 121 °C is statistically significantly different from other extraction techniques.

The ratios of DOC to TDN and TDP, and TDN to TDP in soil extracts were compared among different extraction methods (Table 2). The ratio of DOC to TDN in natural extraction is relatively lower than the autoclave method, ranging between 7.7 and 10.7, and 7.3 and 9.5 for RMFR and AHFR. The ratio of DOC to TDP in natural extraction ranged between 62.2 and 156, and 86.4 and 114 for RMFR and AHFR. Meanwhile, the ratio of TDN to TDP in natural extraction ranged between 5.8 and 18.1, and 11.9 and 14.0 for RMFR and AHFR.

For the autoclaved extraction, DOC ratio to TDN is higher than the natural extraction methods, ranging between 12.5 and 22.8, and 10.1 and 15.5 for RMFR and AHFR, respectively. The DOC ratio to TDP is higher than the natural extraction, ranging between 133 and 247, and 160 and 236 for RMFR and AHFR, respectively. The ratio of TDN to TDP ranged between 6.6 and 13.4, respectively for RMFR. The values are significantly different (*p* < 0.05) using ANOVA and Turkey post-hoc from those in the natural extraction, while the ratio of TDN to TDP for AHFR is higher than in the natural extraction, ranging between 15.2 and 20.3, respectively.

### 2.4. Effect of Ammonia and Nitrate on Microalgae Growth

A significant limiting nutrient in microalgal growth is inorganic nitrogen, such as nitrate or ammonium ion. Figure 4 shows the results obtained from the two microalgae species used in this study, *Nannochloropsis oceanica* and *Chlorella vulgaris*. Results indicating a positive growth effect have been observed in both species based on the recovered ammonia and nitrate from both RMFR and AHFR samples. The highest growth (0.81) is observed in *C. vulgaris* cultured in RM SE media + 121 °C twice. Meanwhile for *N. oceanica*, the highest growth (0.32) is detected in RM SE media + 105 °C. Similar condition of growth is observed in both *N. oceanica* and *C. vulgaris* microalgae in modified SEs of AHFR (Figure 5) where the highest growth (0.42) is detected when *C. vulgaris* is cultured in the AH SE media + 121 °C twice. Meanwhile, the highest growth (0.38) is detected for *N. oceanica* when the species is cultured in the AH SE media + 105 °C. The highest concentration of ammonia in RMFR and AHFR is 2 mg N/L, respectively. While nitrate concentration for both soils are 35 mg N/L and 2.85 mg N/L, respectively. Both parameters are detected when extraction method of 121 °C twice is applied for both soil types. Significant difference is observed (*p* < 0.05) using ANOVA and Turkey post-hoc when the *C. vulgaris* was cultured in RMSE as compared to in AHSE. However, the growth of *N. ocaenica* in both soil extract-containing medium did not exhibit any significant difference (*p* > 0.05) using ANOVA and *t*-test.

### 2.5. Effect of Modified Soil Extract on the Targeted Microalgae Growth

A positive growth pattern was observed in both *N. oceanica* and *C. vulgaris* species under all extraction parameters examined with both AH and RM SEs. *C. vulgaris* growth in RM SE media + 121 °C twice, media + 24 h, media + 105 °C twice and media + 121 °C is higher compared to the control and media + 105 °C (Figure 6A). The statistical analysis showed significant difference at *p* < 0.05 using ANOVA and Turkey post-hoc compared to the control medium. Although the growth of *C. vulgaris* in AH SE media + 121 °C twice is 14% higher (0.42) compared to the control (0.36) (Figure 6B) at similar cultivation day (Day 9), it did not exhibit any significant difference (*p* > 0.05) using ANOVA and *t*-test. In RM SE, *N. oceanica* has recorded a rise in the OD value up to 0.5, indicating a higher growth rate in the control experiment (Figure 7A), exhibiting non-significant results (*p* > 0.05) using ANOVA and t-test. The growth of *N. oceanica* in the control experiment is lower (0.044) compared to all five extraction treatments of modified AH SE (Figure 7B). The negative OD value was remarked in this study. The results are significant at *p* < 0.0.5 using ANOVA and Turkey pos-hoc for all the treatments tested compared to the control.

Meanwhile, the maximum OD value observed in both *N. oceanica* and *C. vulgaris* microalgae are different between the control and modified SEs (Table 3). The maximum OD values of *C. vulgaris*, 0.81 and 0.42, are noted in media + 121 °C twice in both RM SE and AH SE. Meanwhile, the maximum OD value of *N. oceanica*, 0.49 and 0.32, are observed in the control experiment for RM SE and media + 105 °C for AH SE. Statistical analysis showed significant result (*p* < 0.05) using ANOVA and Turkey post-hoc between the control and microalgae culture in RM SE. Similar findings also indicate that AH SE is significant (*p* < 0.05) using ANOVA and Turkey post-hoc at the maximum OD values observed. Further findings exhibited that higher growth of C. *vulgaris* compared to *N. oceanica.* [23] reported that *C. vulgaris* showed higher nitrogen requirements for growth. Study by [24] reported that modified soil extract from sludge pond did influence microalgae growth *C. vulagaris* and *N. oceanica* compared to the control. Although results were presented using optical density, correlation between optical density and number of cells is presented in Appendix A.

The specific growth rate (SGR, (µ) of both microalgae in modified RM and AH SE differs according to the type of media parameterization and extraction technique (Figure 8). The highest SGR value observed in modified AH SE is 0.24 d^−1^ in media + 121 °C for *C.vulgaris* while in modified RM SE is 0.15 d^−1^ in treatment media + 24 h. While *N. oceanica* exhibits the highest SGR, 0.10 d^−1^ and 0.22 d^−1^ in media + 105 °C and in treatment media + 24 h in RM SE and AH SE, respectively. The SGR of *N. oceanica* and *C. vulgaris* are significantly different (*p* < 0.05) in different RM SE and AH SE extraction methods.

The division rate (*k*) of both microalgae in modified RM SE and AH SE are based on the SGR of the microalgae (Table 4). *C. vulgaris* exhibits the highest division rate of 0.21 d^−1^ in media + 24 h for both RM and AH SE. This is followed by 121 °C twice with 0.19 d^−1^. Meanwhile for *N. oceanica*, the highest division rates (k) of, 0.17 d^−1^ and 0.14 d^−1^ are observed in the control medium and at 105 °C extraction followed by medium with 105 °C twice extraction.

## 3. Discussion

### 3.1. Total Dissolved Nitrogen (TDN), Total Dissolved Phosphorus (TDP), and Dissolved Organic Carbon (DOC) Concentrations in Soil Extracts with Natural Extractions

Although microalgae biomass is widely used, its commercial-scale applications are still limited because of the low yield and high cost of microalgae products. Therefore, the present study was endeavored by selecting the use of undisturbed virgin soils originating from RMFR and AHFR as a growth-promoting factor for microalgae. Effective nutrient medium is the key factor that significantly influences the specific growth rate and the final concentration of microalgae. The main objective of this study was to determine the enhancement effects of soil extracts from the reserved forest on the growth of specific microalgae species using novel microplate incubation techniques. The ultimate enhancement effect depends on the quality of the SE, particularly essential total nutrients such as nitrogen and phosphate.

Dissolved organic matter is a small and reactive fraction of the total soil organic matter and is essential in varying biotechnology chemicals [25]. Its movement through soil pores and interaction with solid organic matter renders it a highly dynamic carbon pool subject to physical, chemical, and biological alteration. The recovered TDN, TDP, and DOC concentrations during natural extractions showed no significant increase in concentrations despite increased extraction time. The decrease in concentrations was even observed after one hour. The increase in concentrations could mainly be due to further degradation of soil organic molecules, while the decrease of concentrations could mainly come from the utilization by microbes. For the natural extraction, the soil was dried at 60 °C for a few days before extraction. Some bacteria have survived and assimilated carbon, nitrogen, and phosphorus, causing a decrease of TDN, TDP, and DOC concentrations in soil extract. However, the contribution is negligible because of the a small change observed. The increase of concentrations was also observed, however the change was likewise very small. Our result showed that compounds were significantly released into the solvent during the first hour of extraction. Any inorganic and organic compounds that may be loosely attached to the soil minerals may be resolved quickly when the soil was soaked to Milli-Q (MQ) water. Since soil weathering or degradation rate is usually slow at low temperature (~28 °C), the significant change might not be detected within a day.

Dissolved organic matter is a small and reactive fraction of the total soil organic matter and is essential in varying biotechnology chemicals [25]. Its movement through soil pores and interaction with solid organic matter render it a highly dynamic carbon pool subject to physical, chemical, and biological alteration. The recovered TDN, TDP, and DOC concentrations during natural extractions showed no significant increase in concentrations despite increased extraction time. The decrease in concentrations was even observed after one hour. The increase in concentrations could mainly be due to further degradation of soil organic molecules, while the decrease of concentrations could mainly come from the utilization by microbes. For the natural extraction, the soil was dried at 60 °C for a few days before extraction. Some bacteria have survived and assimilated carbon, nitrogen, and phosphorus, causing a decrease of TDN, TDP, and DOC concentrations in soil extract. However, the contribution is negligible because of the small change observed. The increase of concentrations was also observed, however the change was likewise very small. Our result showed that compounds were significantly released into the solvent during the first hour of extraction. Any inorganic and organic compounds that may be loosely attached to the soil minerals may be resolved quickly when the soil was soaked to Milli-Q (MQ) water. Since soil weathering or degradation rate is usually slow at low temperature (~28 °C), the significant change might not be detected within a day.

### 3.2. Total Dissolved Nitrogen (TDN), Total Dissolved Phosphorus (TDP), and Dissolved Organic Carbon (DOC) Concentrations in Soil Extracts with Autoclaved Extractions Method

The recovered TDN, TDP, and DOC concentrations were much higher for the autoclaved extraction than the natural extraction. The soil sterilization method has been demonstrated to increase the amount of extract for manganese, ammonium, and organic nitrogen without altering the soil’s physical properties [26]. Apart from the increased extractable Mn, N, P, and S level, the increased level of organic carbon is also observed because of the autoclaving procedure. More compounds and molecules can be released to the solvent, since the heat may provide more energy to break down solvent compounds and molecules. Among the autoclave extraction methods, the recovered TDN, TDP, and DOC concentrations were generally higher at the 121 °C extraction method than the 105 °C extraction and autoclaved twice compared to autoclave just once. The concentrations of the dissolve organic matter (DOM) in soil extract appear to be proportional to the amount of heat energy that the soil received. The energy required to breakdown the molecules into carbon, nitrogen, and phosphorus compounds released to the soil extract, should exceed the activation energy, to allow the breakdown of the chemical bonds [27,28]. It is difficult to break down any chemical bonds at low temperature in natural extraction because the heat energy is too low to exceed the activation energy. Hence, the concentrations of TDN, TDP, and DOC did not increase with the increase of extraction time. This indicated that the amount of heat during autoclave extractions could be the most important factor to increase the yields of elements in soil extracts.

This study examined two soil samples obtained from two different reserved forests (RMFR and AHFR). Our study showed that the concentrations of TDN, TDP and DOC in soil extracts from RMFR were interestingly higher than those from AHFR. The concentrations of soil extracts at RMFR were up to 18.5 times higher than those at AHFR. This suggests that different soil types largely affect the recovered concentrations of carbon, nitrogen, and phosphorus in soil extracts. According to [29], the peat swamp soil collected at Tanjung Karang situated at the edge of RMFR, contained nearly 50% carbon, while the carbon content in soil at AHFR ranged between 0.8 and 1.9% [18]. The high carbon content at RMFR could result in a high recovery of carbon, nitrogen, and phosphorus in soil extracts. Our results suggest that the recovery of soil extracts depends upon the amount of heat provided during extraction and the initial carbon, nitrogen, and phosphorus contents in the studied soils.

The ratios of carbon to nitrogen (C/N ratios) in natural extraction method ranged between 7 and 10. The low C/N in natural extraction method suggests that the compounds loosely attached to soil minerals could be nitrogen-rich, such as proteins and glycoproteins. Moreover, the soils contains a significant numbers of bacteria. The process of soaking dried soil in MQ water could release the dead bacteria. The C/N ratios of bacteria usually range around 4–5 [30]. The relatively low C/N ratios of soil extract during natural extraction could partially come from dead bacteria. For autoclave samples, C/N ratios were elevated to 10–23. It suggests that more carbon-rich compounds could be released to the soil extracts. For example, cellulose is a major plant material, derived from d-glucose units, condensed through β (1→4)-glycosidic bonds. The β-glycosidic bonds are much stronger than α-glycosidic bonds which form starch and glycogen. During autoclave, the heat energy could cleave this strong β (1→4)-glycosidic bonds, allowing more carbon-rich to be released into the soil extracts. Our results suggest that different chemical compounds can be released from different extraction methods. The chemical compositions could be altered using autoclave [31]. Some essential compounds for the growth of organisms can be degraded or altered to no longer be available for the target organisms. Therefore, the soil extracts produced by autoclaving at 121 °C might not be the best because essential compounds for the growth of target organisms could be lost, although the highest recovery of carbon, nitrogen, and phosphorus can be obtained.

A study done by [32], indicates that the highest recovery of total N from soil extract were closed to 100%. Similar phenomenon was obtained in our study with the highest value of 70%–120% TDN concentration recovery obtained at RMFR using 121 °C twice extraction method. This implies that soil extraction condition significantly affected the recovery of nitrogen. Moreover, [33] in their study has reported TNP recovery are 39% to 97%. DOC studied by [34] implies a higher recovery accounted for almost 58.8–84.5% of the total DOC.

### 3.3. Effect of Ammonia and Nitrate on the Growth of Microalgae

Microalgae can assimilate nitrogen from a variety of sources [21,35,36]. Ammonia, nitrite, nitrate, and many dissolved organic nitrogen (urea, free amino-acids and peptides) are regarded as the main nitrogen sources for microalgae [37,38,39]. Ammonia, urea, and nitrates are often selected as the nitrogen source for mass cultivation of microalgae [40,41]. The interaction of nitrogen sources on nitrogen assimilation by microalgae varies among species and is strongly influenced by the environmental condition [42,43,44,45,46].

In this study, the presence of ammonia directly interferes with the nitrate uptake by both *N. oceanica* and *C. vulgaris*. The microalgae preferred ammonia as its nitrogen source where his phenomenon can be monitored in the Chlorella sp. *Chlorella* has developed an alternative adaptation strategy to acquire N from endosymbiotic relations with *Paramecium bursaria* in the natural environment and can use ammonium or amino acids but not nitrate or nitrite [47]. The preferential uptake of ammonia and the suppression of nitrate uptake by ammonia availability have been previously studied [48,49,50]. The low effect of nitrate on the growth rate could be due to the inactivation of nitrate reductase system by ammonia [48,49,50,51] or by the by-product of ammonia assimilation. Ammonia requires no enzymatic reduction for assimilation, but nitrate must be reduced to ammonia before the microalgae can assimilate it. The reduction of nitrate to ammonia involves two independent enzymatic steps. First, the reduction of nitrate to nitrite catalyzed by NADH_2_-nitrate reductase, and second, the reduction of nitrite to ammonia catalyzed by the ferredoxin-nitrite reductase. There is a considerable energy requirement for utilizing nitrate because of the number of electrons required to reduce nitrate to ammonia (Figure 9).

Ammonia and nitrate exhibit different effects on the biomass growth of both species. The result showed that the uptake rate of ammonia was higher than nitrate. Our finding was in line with the research done by [52], where *Nannochloropsis* sp. produced different cell densities growing on both ammonia and nitrate. However, a high cell density was reached more rapidly when it grew on ammonia than nitrate. Similar results were reported by the study done by [53], where dry weight and growth rate of *C. vulgaris* increased in the presence of ammonia and subsequently, lower growth in nitrate. Most microalgae convert nitrate ions to ammonium ion inside the algal cell, which requires energy before amino acid synthesis [54]. Thus, the lower growth in nitrate presence could be due to other intracellular processes to convert nitrate ions into ammonium ions. Indeed, higher growth rates were observed for ammonia nitrogen utilization which could be directly assimilated.

As studied by [55], for *Ellipsoidion sp*., ammonium has been demonstrated to produce higher biomass than those of urea and nitrate. Similar phenomenon were observed with *N. oculata* in which ammonium-*N* (3.73 mg ammonium·L^−1^) was taken up by microalgae, as reported by [56]. Moreover, the positive effect of ammonium on significantly enhanced biomass production was highlighted by [57] with the ammonium concentration of 2 mg NH_3_/L, However, in contrast, *Chlorella sp*. and *Neochloris oleoabundans* had high biomass concentration when urea and nitrate were utilized as nitrogen source as reported by [58].

### 3.4. Effect of Modified Soil Extract on the Targeted Microalgae Growth

The soil has an oligotrophic environment; it contains all the necessary elements for the optimum growth of microalgae when cultivated in an artificial medium [59]. Soil contains many free enzymes that play a critical role in catalyzing the reactions leading to the decomposition of organic matter that stimulate microalgal growth [60]. The combination of soil extract and commercial media might be used as an alternative method to produce microalgae, leading to cost reduction instead of solely relying on commercial media. A study by [61] revealed that the combination of culture medium with vitamins and soil extract led to a significant increase in the optical density of *C. vulgaris*. Therefore, this variant of the culture medium is the most effective to obtain large volumes of algae suspension.

In the present study, Ayer Hitam soil extract was believed to enhance microalgal growth proportionate to Raja Musa soil extract. This phenomenon could be due to the difference in organic matter content. Nevertheless, the growth pattern of both microalgae in this study was similar with different extraction parameters of control and SEs. Our findings were in line with the study by [62,63,64] where the control and SE treatment displayed a similar growth pattern. However, in some cases, microalgae, due to high nutrient levels coupled with unique extraction methods, grew well in different extraction parameters compared to the control. This could be due to specific extraction parameters reduce and precipitate the necessary vitamins and minerals present in the treated SE, and the N and P concentrations [65,66].

The maximum OD observed in RM SE was higher than AH SE for both microalgae species. More specifically, the maximum OD of *N. oceanica*, and *C. vulgaris*, were higher in RM SE than AH SE due to the high TDN and TDP content obtained from the present study. Both microalgae showed better growth trends because of soil extract, which increased the nitrogen content in the culture medium, thus promoting the growth of microalgae. According to [64] the presence of manganese in the soil extracts is necessary to facilitate the microalgae growth. A previous study by [4] highlighted that soil extracts in culture medium promoted the microalgae growth rate. In the present study, higher TDN and TDP content increased the microalgae growth. As for modified SEs, media + 105 °C and media + 121 °C twice have stated high OD values. As stated by [65], autoclaving at high temperatures for a prolonged time could kill bacterial and protozoal, which could inhibit microalgae growth. Studies on the drastic increase in dissolved organic matters after autoclaving have been reported by [67,68,69].

A study by [62] observed that lower optical density for *Chlorella vulgaris* was shown when 30 mL/L of the soil extract concentration was used. The exponential growth phase lasted longer with the maximum optical density, 0.561 was observed on the 5th day of the cultivation period. This indicates a lowering of about 60% optical density compared to that from our observation using RM SE. In a similar manner, a study by [61], indicates that the maximum growth of *Chlorella vulgaris* was reached on the 12th day of cultivation with optical density value of 0.152. According to [4], *Nannochloropsis* sp. growth cultivation was increased from day 1 to 16 with highest optical density recorded of 0.1557. After day 6, *Nannochloropsis sp*. entered the stationary phase because of the accumulation of waste and nutrient insufficiency. However, in the present study, maximum OD for *Nannochloropsis oceanica* was recorded at 0.42 with the media + 105 °C extraction. Results obtain implies that the highest growth rate has been achieved using additional soil extracts.

Based on the types of soil extracts, the method of enhancement or extraction, and the culture media used in the control experiment, the SGR of microalgal species were ranged. The SGR of *C. vulgaris* was relatively higher than the control in both RM and AH SEs (Figure 8). The relatively higher SGR indicates that the addition of extraction parameters and nutrients in the modified SE are essential to the exponential rise. Higher SGR value in control experiments by *N. oceanica* of RM SE indicated species dependence and variability. RM SE has a relatively higher nutrient content, affecting the SGR value of both species to the maximum levels attained in RM SE. This could be due to the basal medium that contains nitrogen mainly in the form of NO^3−^ which is reduced to ammonia during the cultivation period. Soil extract also contains ammonium (NH^4+^). These high levels of environmental ammonium inhibit cell growth resulting in the imbalance of ammonia diffusion across the plasma membrane, thus, resulting in additional support on microalgae production and no increase of total biomass concentrations over time. Besides, the utilization of microplate incubation technique in quantifying the microalgae growth might be due to low volume (micro-well) with high microbial biomass concentration [69,70,71,72].

A study by [73] on the investigation for effect of different carbon sources in soil extract medium for the growth of *C. vulgaris* indicated that 1 g/L glucose concentration, enhanced the production of biomass up to 1.23 g/L. This indicates the role of soil extract in enhancing the growth of *C. vulgaris*. Moreover, [4] has reported that using microelement from soil extracts will be beneficial for the cultivation of microalgae. Based on his finding, soil extracts initiate rapid reproductive and initiate rapid growth of certain microalgae such as *Ulva* (50%), *Dictyota* (42%), and *Pterocladia* (47.5%). Cultivation of *Nannochloropsis* sp. in 16 days incubation with variety of soil extracts concentration (3.5% to 7.0%) resulted in higher growth rates, where 7.0% soil extracts concentration produce higher biomass. Ref. [74] reported an increase in growth rate, cell number, and doubling time of *Cosmarium subtumidum* in the cultivation with soil extract for up to 2628.47 cell/mL. The increase in the number of cells has been influenced by the increase in the concentration of soil extracts.

The selected RMFR soils were tested in the outdoor cultivation for mass production of microalgae using photobioreactor. Higher density of microalgae produced are useful in various industries especially in aquaculture, cosmetic, energy, and biomedical use. This will contribute to generating lower cost product.

## 4. Materials and Methods

### 4.1. Sample Collection and Preparation

Soil samples were collected from two different locations, Raja Musa Forest Reserve (RMFR) (3°26′45.2″ N 101°19′20.9″ E) and Ayer Hitam Forest Reserve (AHFR) (3°00′27.7′′ N 101°38′46.9′′ E), Selangor Malaysia (Appendix A). RMFR is a peat swamp forest, while AHFR is categorized as a lowland dipterocarp forest. RMFR is located northwest of Kuala Lumpur and has an area of 23,486 hectares. Meanwhile, AHFR is located south of Kuala Lumpur and has an area of 1248 hectares. AHFR is currently under the sustainable management of Universiti Putra Malaysia (UPM). Soil surface or O horizon layer was removed, and the subsurface soil was collected using random sampling technique according to the United State Department of Agriculture (USDA) methods [75]. Each sample was collected from at least three sampling sites within 10 m away at each site. The coarse materials such as stones, woods, and roots were removed before the samples were mixed. After sampling, the soil samples were immediately brought back to the lab and stored frozen until the next procedure. The frozen samples were slowly thawed and dried at 60 °C in a drying oven DKM-400 (Yamato Scientific Co., Ltd, Tokyo, Japan) for a few days until completely dried. The dried soil samples were ground into fine particles. Then, the samples were sieved to remove non-soil materials. The dried fine soils was stored in the dark at room temperature until soil extraction.

### 4.2. Soil Extraction

The soil extraction was conducted using the fine soil samples using a modified method [31]. Various aqueous extraction methods were tested as described in Table 5. For each method, 20 g of dried soil was added to 200 mL (MQ) water in a 400 mL centrifuge bottle. For natural extraction, the samples were placed in the dark at room temperature for 1, 4, and 24 h, respectively. For autoclaved extraction, the samples were autoclaved using autoclave SX-500 (Tomy Seiko Co., Ltd, Tokyo, Japan) separately for (1) 105 °C, (2) 121 °C, (3) 105 °C after 24 h natural extraction, (4) 121 °C after 24 h natural extraction, (5) 105 °C twice, and (6) 121 °C twice. The autoclave duration was set for one hour, and the procedure of autoclave twice treatments comprised the sample that autoclaved for one hour, cooled and autoclaved again for one hour. The interval between two autoclave cycles was about 30 min. After autoclaving or natural extraction, the samples were centrifuged at 2500 rpm for 15 min using Beckman centrifuge Allegra X-30R (Beckman Coulter, Indianapolis, IN, USA). Then, the supernatants were carefully taken and filtered by glass fiber filter (Whatman GF/F). The filtered water samples were stored in Revco, ULT-390-10 freezer at −20 °C (Thermo Fisher Scientific, Japan) until further analyses.

### 4.3. Sample Analyses

The concentrations of total dissolved nitrogen (TDN) and total dissolved phosphorus (TDP) were determined using Lovibond MD 600D (The Tintometer Limited, Amesbury, UK), a portable spectrophotometer. The sample was added to a vial with designated reagents or the designated reagent was added to the vial after the sample was added. Dissolved organic carbon (DOC) was measured using Shimadzu TOC-L CSH (Shimadzu Corp., Kyoto, Japan). The measurements were conducted three or four times for each sample to calculate average values and standard deviation.

### 4.4. Microalgae

The target microalgal species used in this study were *Chlorella vulgaris* (TRG 4C) and *Nannochloropsis oceanica* (TRG 3A) obtained from the National Institute for Environmental Studies (NIES), Japan. Conway media was prepared from five basic solutions as described by [76] comprising the mineral solution −100 g of NaNo_3_, 45 g of disodium EDTA (C_6_H_16_N_2_O_8_), 33.6 g of H_3_BO_3_, 20 g of NaH_2_PO_4_·4H_2_O, 1.3 g of FeCl_3_·6H_2_O, 0.36 g of MnCl_2_·4H_2_O, and 1 mL trace metal solution in 1 L of MQ water; trace metal solution—0.21 g of ZnCl_2_, 0.2 g of CoCl_3_·6H_2_O, 0.09 g of (NH_4_)_6_MO_7_O_2_·4H_2_O, and 0.2 g of CuSO_4_·5H_2_O in 100 mL MQ water; vitamin solution—0.2 g of thiamine (B1), cyanocobalamin (B12) in 100 mL of MQ water; silicate solution—2 g of Na_2_SiO_3_ in 100 mL of MQ water; and nitrate solution—2 g of KNO_3_ in 100 mL of MQ water. The media was prepared by adding 1 mL of main mineral, silicate, and nitrate solution to the Schott bottle to prepare 1 L volume media. After autoclaving the prepared media, 1 mL of NH_4_Cl and vitamin solution were added into the cooled medium to give a final medium concentration of 5.0 × 10^−4^ M. The cultures were grown at 25 ± 0.5 °C under a light intensity of 33.75 µmol photons m^−2^·s^−1^ on a 12 h light:12 h dark cycle. The stock cultures were acclimatized to the experimental conditions prior to the experiment before the strains were tested on sludge extracts.

Microplate-incubation was performed for the microalgae species in the five different extraction parameters of SE using 96-well microplates (Figure 10). Each well in the microplate can be filled up to 200 µL of solution. The border wells of the microplate were filled with 200 µL MQ water to prevent evaporation during the study. Previous studies showed that the border wells of the microplates were not used during experiments as it expose the wells to strong air currents, although microalgae growing in the border wells have more access to light and CO_2_ [77,78,79]. The remaining wells were filled with 195 µL of suitable media + 5 µL of 105 °C SE in the 2nd column (blank), and the 3rd column filled with 175 µL of suitable media + 5 µL of 105 °C SE + 20 µL of microalgae (experiment), as shown in Figure 10 to record the exponential phase of microalgae. The same steps were repeated in the 4th to 11th columns of a microplate with 105 °C twice (Column 5), 121 °C (Column 7), 121 °C twice (Column 9), and 24 h natural extraction (Column 11). For the control experiment, (Conway media) without SE was tested with microalgae in another microplate. The microplates were sealed with parafilm after pipetting all the wells in the microplate to prevent evaporation by preserving the air humidity in the microplate wells and preventing external contamination before incubation. The microplates were incubated for nine days, and the biomass or growth of microalgae was determined by optical density (OD) at 680 nm for every 24 h using the microplate reader Infinite M200 PRO (Tecan, Austria). For every 24 h of OD measurement, each one of the wells containing controls and samples was mixed using 8-channel Eppendorf pipettor before measuring the OD to mix the microalgae suspended in the bottom well with the solution.

### 4.5. Data Analysis

Three microplate replicates for each control and sample in a column were tested. The OD of the control and sample was subtracted to get the net OD mean value. OD measurements were used in this study to determine the microalgal biomass as it is simple, fast, and a commonly used technique to measure the algal culture density [80,81,82]. The specific growth rate (µ) and the division rate (*k*) of microalgae were calculated as follows,
(1)$$µ=\frac{\ln\left(N_{2}-N_{1}\right)}{\left(t_{2}-t_{1}\right)}$$
(2)$$k\;=\;\frac{µ}{ln2}$$
where *N*_2_ and *N*_1_ are the OD at times *t*_2_ and *t*_1_ respectively.

The TDN and TDP content, growth of microalgae, and maximum OD in respective temperature treatment parameters of RM and AH SE were analyzed using independent samples t-test, one-way analysis of variance (ANOVA), and Tukey post-hoc analysis. Significant differences between the different extraction parameters were calculated at 95% confidence interval level. All statistical analyses were conducted using IBM SPSS (Statistical Package for the Social Sciences) statistics 20 software. The test conducted for statistical analysis is shown in detailed in the Appendix A

## 5. Conclusions

In conclusion, the analysis showed the promising result of enhanced microalgae (marine and freshwater) growth with additional enrichment from treated forest soil extracts. Potential microalgae enriched biomass can be increased economically using soil extract as an alternative to expensive vitamins. The result of any enrichment experiment and possible future application to mass culture will be evaluated by the consistency of the soil extract and the species of microalgae being studied. The study shows that autoclaving helps to recover nutrients at elevated temperatures and potentially inhibits pathogens. The results suggested that the best extraction methods for the highest recovery of dissolved organic carbon (DOC), total dissolved nitrogen (TDN), and total dissolved phosphorus (TDP) are the autoclaved method at 121 °C twice. However, most soil extract sources from RM enhanced microalgae growth up to 54.3% than AH (14%). A similar finding was observed for CNP recovery where increasing rate is 87%, 84%, and 95%. The medium of artificial culture alone is costly and insufficient; therefore, this study successfully discovered natural growth-promoting substances from virgin forest soil extract proven to increase the growth of microalgae. This is an important finding for the large-scale cultivation and can help the farmers in aquaculture industry as well. As per our knowledge, industries related to microalgae are now expanding and researchers are trying hard to seek economical raw materials that can provide optimum results. Hence the findings from this study would benefit the microalgae production industry in reducing the cost of mass crop enrichment with soil extracts and simultaneously could also increase the growth and nutritional value of microalgae. Metagenomics research suggested to reveal the functional and metabolic diversity of microbial communities in soil that helps in increasing the growth of microalgae.

## Figures and Tables

**Figure 1 molecules-26-00653-f001:**
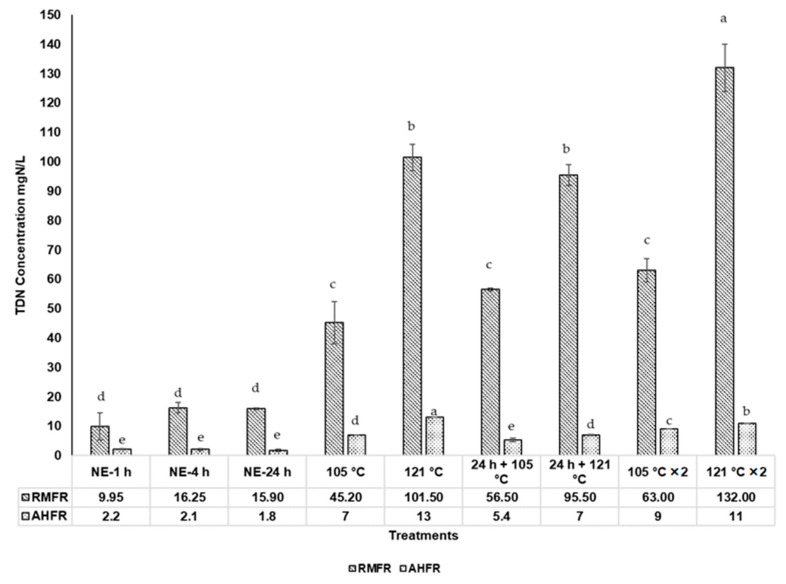
Total dissolved nitrogen (TDN) concentrations in soil extract from different extraction methods; Raja Musa Forest Reserve (RMFR) and Ayer Hitam Forest Reserve (AHFR) for nine different extraction methods; (NE-1 h-soil extracted at room temperature 1 h; NE-4 h-soil extracted at room temperature 4 h; NE-24 h-soil extracted at room temperature 24 h; 105 °C—autoclave 105 °C; 121 °C—autoclave 121 °C; 24 h + 105 °C—autoclave 105 °C after natural extraction 24 h; 24 h + 121 °C—autoclave 121 °C after natural extraction 24 h; 105 °C × 2—autoclave 105 °C twice; 121 °C × 2—autoclave 121 °C twice. Error bars represent standard deviation (*n* = 3). ^a–e^ Mean value with different superscripts are significant different (*p* < 0.05) using ANOVA and Turkey post-hoc.

**Figure 2 molecules-26-00653-f002:**
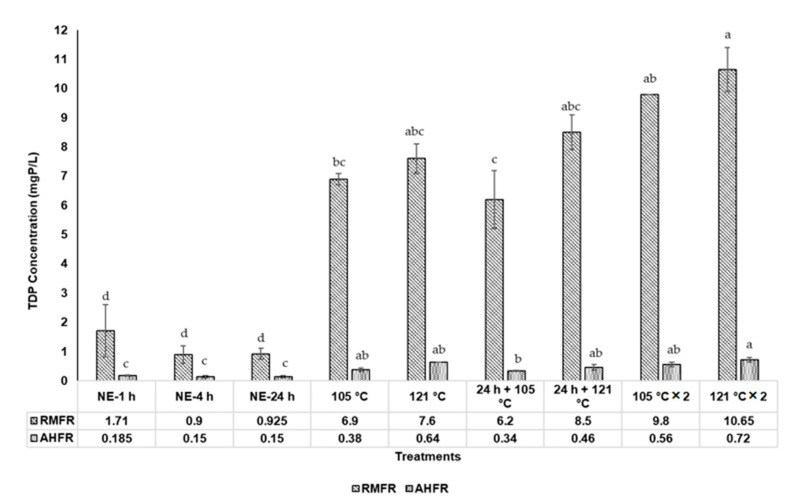
Total dissolved phosphorus (TDP) concentrations in soil extract from different extraction methods; Raja Musa Forest Reserve (RMFR) and Ayer Hitam Forest Reserve (AHFR) for nine different extraction methods; (NE-1 h-soil extracted at room temperature 1 h; NE-4 h-soil extracted at room temperature 4 h; NE-24 h-soil extracted at room temperature 24 h; 105 °C—autoclave 105 °C; 121 °C—autoclave 121 °C; 24 h + 105 °C—autoclave 105 °C after natural extraction 24 h; 24 h + 121 °C—autoclave 121°C after natural extraction 24 h; 105 °C × 2—autoclave 105 °C twice; 121 °C × 2—autoclave 121 °C twice. Error bars represent standard deviation (*n* = 3). ^a–d^ Mean value with different superscripts are significant different (*p* < 0.05) using ANOVA and Turkey post-hoc.

**Figure 3 molecules-26-00653-f003:**
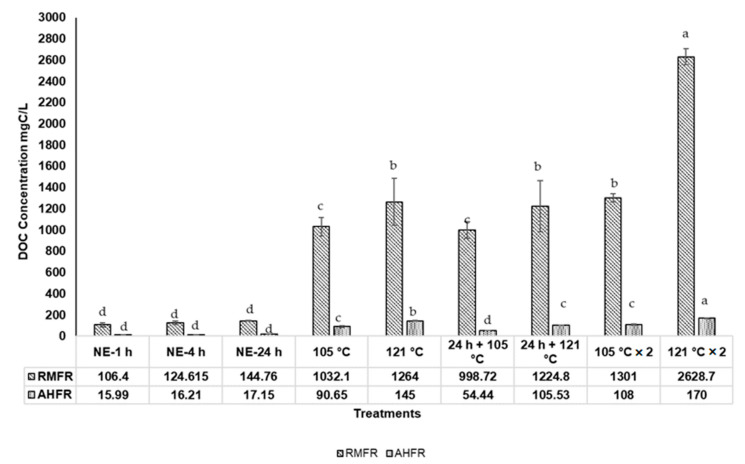
Dissolved organic carbon (DOC) in soil extract from different extraction methods; Raja Musa Forest Reserve (RMFR) and Ayer Hitam Forest Reserve (AHFR) for nine different extraction methods; (NE-1 h-soil extracted at room temperature 1 h; NE-4 h-soil extracted at room temperature 4 h; NE-24 h-soil extracted at room temperature 24 h; 105 °C—autoclave 105 °C; 121 °C —autoclave 121 °C; 24 h + 105 °C—autoclave 105 °C after natural extraction 24 h; 24 h + 121 °C—autoclave 121°C after natural extraction 24 h; 105 °C × 2—autoclave 105 °C twice; 121 °C × 2—autoclave 121 °C twice. Error bars represent standard deviation (*n* = 3). ^a–d^Mean value with different superscripts are significant different (*p* < 0.05) using ANOVA and Turkey post-hoc.

**Figure 4 molecules-26-00653-f004:**
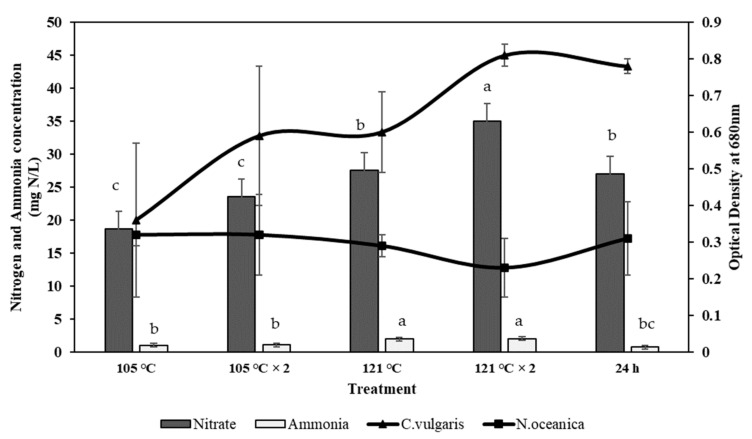
Effect of ammonia and nitrate concentration in on the growth of *C. vulgaris* and *N. oceanica* from Raja Musa Forest Reserve (RMFR) with different extraction methods. (105 °C—autoclave 105 °C; 105 °C × 2—autoclave 105 °C 2 × after natural extraction 24 h; 121 °C—autoclave 121 °C after natural extraction 24 h; 121 °C × 2—autoclave 121 °C twice after natural extraction 24 h; 24 h—natural extraction for 24 h). Error bars represent standard deviation (*n* = 3). ^a–c^ Mean value with different superscripts are significant different (*p* < 0.05) using ANOVA and Turkey post-hoc.

**Figure 5 molecules-26-00653-f005:**
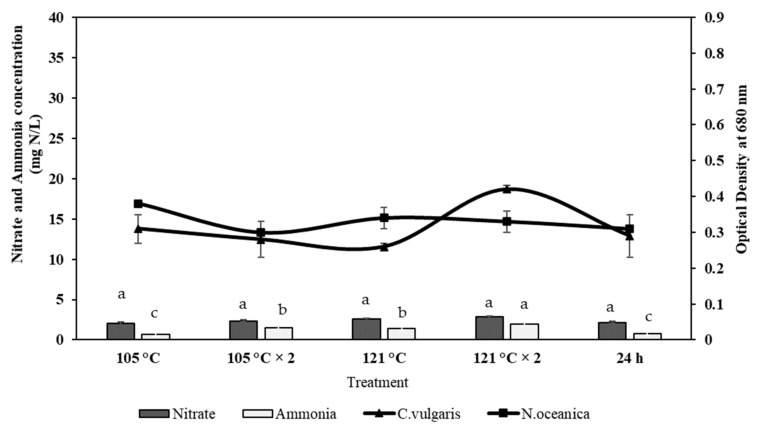
Effect of ammonia and nitrate concentration in on the growth of *C. vulgaris* and *N. oceanica* from Ayer Hitam Forest Reserve (AHFR) with different extraction methods. (105 °C—autoclave 105 °C; 105 °C × 2—autoclave 105 °C × 2 after natural extraction 24 h; 121 °C—autoclave 121 °C after natural extraction 24 h; 121 °C × 2—autoclave 121 °C twice after natural extraction 24 h; 24 h—natural extraction for 24 h). Error bars represent standard deviation (*n* = 3). ^a–c^ Mean value with different superscripts are significant different (*p* < 0.05) using ANOVA and Turkey post-hoc.

**Figure 6 molecules-26-00653-f006:**
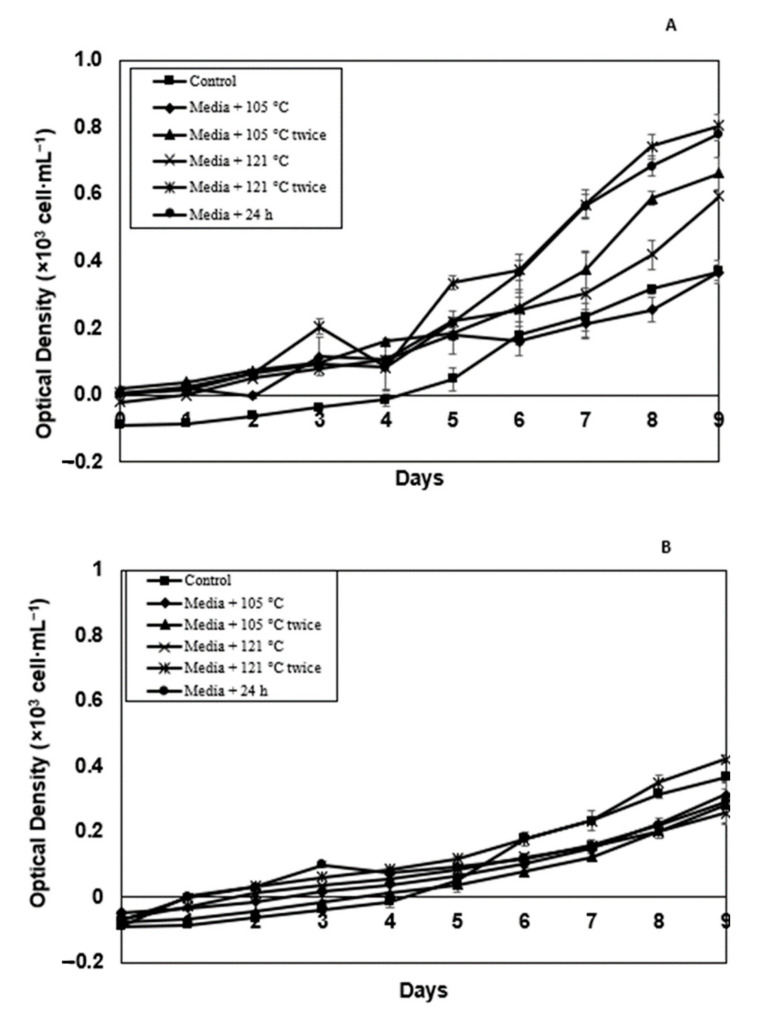
Optical density at 680 nm of *C. vulgaris* in control, media + 105 °C, media + 105 °C twice, media + 121 °C, media + 121 °C twice and media + 24 h (media + 105 °C—autoclave 105 °C; media + 105 °C twice—autoclave 105 °C 2 × after natural extraction 24 h; media + 121 °C—autoclave 121 °C after natural extraction 24 h; media + 121 °C × 2—autoclave 121 °C twice after natural extraction 24 h; media + 24 H—natural extraction for 24 h). (**A**) RM SE and (**B**) AH SE. Error bars represent standard deviation (*n* = 3).

**Figure 7 molecules-26-00653-f007:**
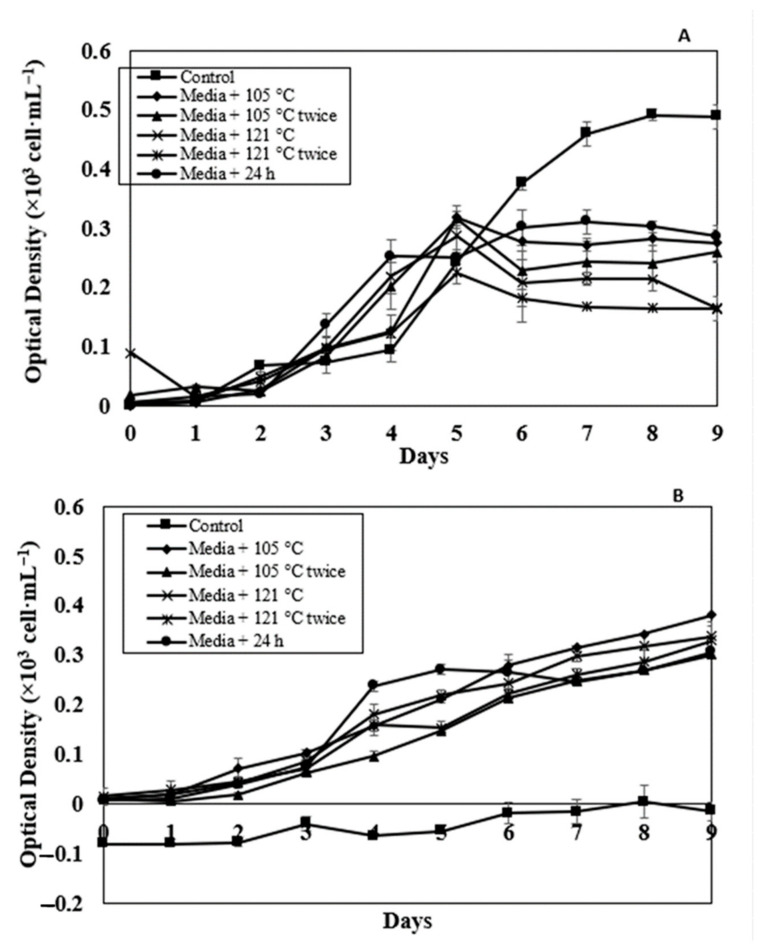
Optical density at 680 nm of *N.oceanica* in control, media + 105 °C, media + 105 °C twice, media + 121 °C, media + 121 °C twice and media + 24 h (media + 105 °C—autoclave 105 °C; media + 105 °C twice—autoclave 105 °C 2 × after natural extraction 24 h; media + 121 °C—autoclave 121°C after natural extraction 24 h; media + 121 °C × 2—autoclave 121 °C twice after natural extraction 24 h; media + 24 H—natural extraction for 24 h). (**A**) RM SE and (**B**) AH SE. Error bars represent standard deviation (*n* = 3).

**Figure 8 molecules-26-00653-f008:**
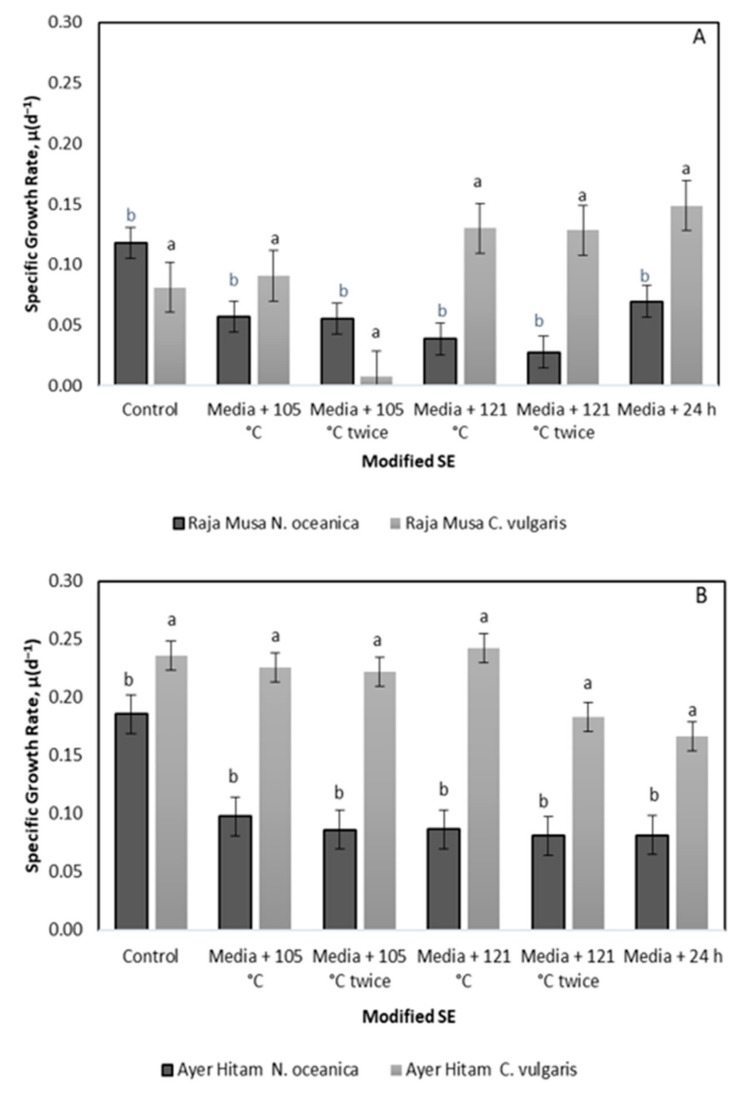
Specific growth rate, µ of *N. oceanica, C. vulgaris,* in control, media + 105 °C, media + 105 °C twice, media + 121 °C, media + 121 °C twice and media + 24 h at (**A**) RM SE and (**B**) AH SE. (media + 105 °C—autoclave 105 °C; media + 105 °C twice—autoclave 105 °C 2 × after natural extraction 24 h; media + 121 °C—autoclave 121 °C after natural extraction 24 h; media + 121 °C × 2—autoclave 121 °C twice after natural extraction 24 h; media + 24 h—natural extraction for 24 h). Error bars represent standard deviation (*n* = 3). ANOVA shows significant difference at *p* < 0.05. Values shown are mean of three replicates with ^+^ SD. ^a,b^ Mean value in same row with different superscripts are significantly different (*p* < 0.05) using ANOVA and post-hoc.

**Figure 9 molecules-26-00653-f009:**
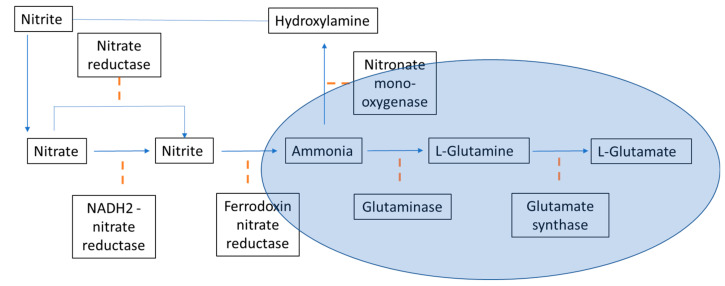
Nitrogen metabolism reduction and fixation in photosynthetic organisms. The blue oval depicts the pathway that has been used by microalgae in order to acquire N directly from ammonium.

**Figure 10 molecules-26-00653-f010:**
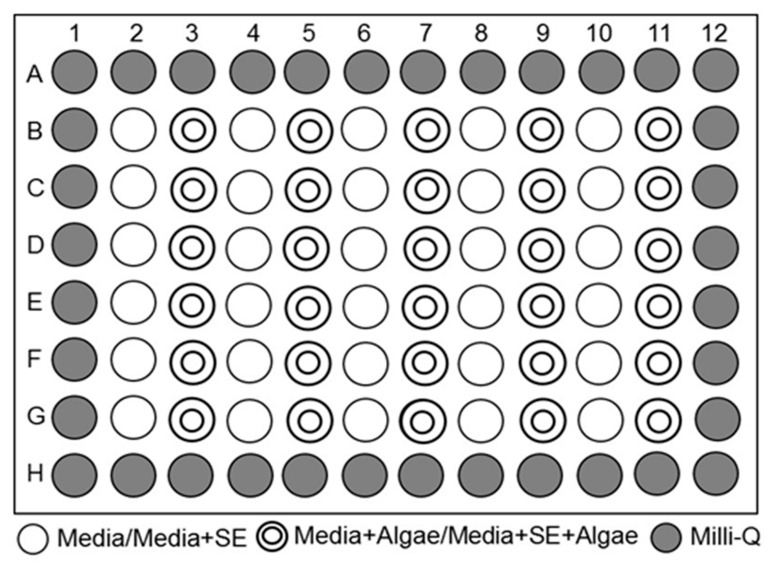
Microplate-incubation technique.

**Table 1 molecules-26-00653-t001:** Ratios of concentrations of TDN, TDP, and DOC in Raja Musa Forest Reserve (RMFR) to Ayer Hitam. Forest Reserve (AHFR).

Extraction Methods	RMFR/AHFR		
	N Ratio	P Ratio	C Ratio
NE-1 h	4.7 ± 0.02 ^c^	9.2 ± 0.01 ^a^	6.7 ± 0.04 ^d^
NE-4 h	7.7 ± 0.04 ^abc^	6.0 ± 0.03 ^a^	7.7 ± 0.04 ^d^
NE-24 h	9.2 ± 0.06 ^abc^	6.2 ± 0.02 ^a^	8.4 ± 0.09 ^d^
105 °C	6.5 ± 0.01 ^ab^	18.2 ± 0.08 ^a^	11.4 ± 0.07 ^c^
121 °C	7.8 ± 0.08 ^abc^	11.9 ± 0.02 ^a^	8.7 ± 0.04 ^d^
NE + 105 °C	10.5 ± 0.02 ^abc^	18.2 ± 0.02 ^a^	18.3 ± 0.08 ^a^
NE + 121 °C	13.6 ± 0.07 ^a^	19.7 ± 0.08 ^a^	11.6 ± 0.02 ^d^
105 °C × 2	7.0 ± 0.05 ^abc^	17.5 ± 0.09 ^a^	12.0 ± 0.05 ^d^
121 °C × 2	12 ± 0.04 ^ab^	9.2 ± 0.03 ^a^	15.46 ± 0.07 ^b^

Values shown are mean of three replicates with ± SD. ^a–d^ Mean values in same row with different superscripts are significantly different (*p* < 0.05) using ANOVA and Turkey post-hoc.

**Table 2 molecules-26-00653-t002:** Ratios of DOC to TDN and TDP, and TDN to TDP for RMFR and AHFR using different extraction methods.

	Raja Musa Forest Reserve (RMFR)			Ayer Hitam Forest Reserve (AHFR)		
	C/N	C/P	N/P	C/N	C/P	N/P
NE-1 h	10.7 ± 0.08 ^ab^	62.2 ± 0.05 ^b^	5.8 ± 0.04 ^a^	7.3 ± 0.03 ^b^	86.4 ± 0.01 ^a^	11.9 ± 0.03 ^a^
NE-4 h	7.7 ± 0.08 ^b^	138.5 ± 0.05 ^ab^	18.1 ± 0.09 ^a^	7.7 ± 0.016 ^b^	108.1 ± 0.02 ^a^	14.0 ± 0.03 ^a^
NE-24 h	9.1 ± 0.01 ^b^	156.5 ± 0.06 ^ab^	17.2 ± 0.09 ^a^	9.5 ± 0.09 ^ab^	114.3 ± 0.02 ^a^	12.0 ± 0.06 ^a^
105 °C	22.8 ± 0.04 ^a^	149.6 ± 0.04 ^ab^	6.6 ± 0.05 ^a^	13.0 ± 0.07 ^ab^	238.6 ± 0.07 ^a^	18.4 ± 0.04 ^a^
121 °C	12.5 ± 0.06 ^ab^	166.3 ± 0.02 ^ab^	13.4 ± 0.01 ^a^	11.2 ± 0.01 ^ab^	226.6 ± 0.07 ^a^	20.3 ± 0.09 ^a^
NE + 105 °C	17.7 ± 0.01 ^ab^	161.1 ± 0.01 ^ab^	9.1 ± 0.01 ^a^	10.1 ± 0.02 ^ab^	160.1 ± 0.04 ^a^	15.9 ± 0.01 ^a^
NE + 121 °C	12.8 ± 0.02 ^ab^	144.1 ± 0.01 ^ab^	11.2 ± 0.02 ^a^	15.1 ± 0.04 ^a^	229.4 ± 0.08 ^a^	15.2 ± 0.07 ^a^
105 °C × 2	19.1 ± 0.07^ab^	132.8 ± 0.00 ^ab^	6.9 ± 0.07 ^a^	12.0 ± 0.06 ^ab^	192.9 ± 0.09 ^a^	16.1 ± 0.02 ^a^
121 °C × 2	19.9 ± 0.04 ^ab^	246.8 ± 0.05 ^a^	12.4 ± 0.09 ^a^	15.5 ± 0.01 ^a^	236.1 ± 0.02 ^a^	15.3 ± 0.01 ^a^

Values shown are mean of three replicates with ± SD. ^a,b^ Mean values in same row with different superscripts are significantly different (*p* < 0.05) using ANOVA and Turkey post-hoc.

**Table 3 molecules-26-00653-t003:** The maximum OD of *N. oceanica, C. vulgaris* on control, 105 °C, 105 °C twice, 121 °C, 121 °C twice and 24 hours’ soil extraction (SE) from *Raja Musa* Forest Reserve (RM) and *Ayer Hitam* Forest Reserve (AH).

Types of SE	Microalgae	Control			Modified SE		
			Media + 105 °C	Media + 105 °C Twice	Media + 121 °C	Media + 121 °C Twice	Media + 24 h
**RM SE**	*N.oceanica*	0.49 ± 0.03 ^a^	0.32 ± 0.05 ^b^	0.32 ± 0.11 ^b^	0.29 ± 0.03 ^b^	0.23 ± 0.08 ^b^	0.31 ± 0.10 ^b^
	*C. vulgaris*	0.37 ± 0.03 ^c^	0.36 ± 0.21 ^c^	0.59 ± 0.19 ^b^	0.60 ± 0.11 ^b^	0.81 ± 0.03 ^a^	0.78 ± 0.02 ^a^
**AH SE**	*N.oceanica*	0.00 ± 0.03 ^c^	0.38 ± 0.00 ^b^	0.30 ± 0.00 ^b^	0.34 ± 0.03 ^b^	0.33 ± 0.03 ^b^	0.31 ± 0.00 ^b^
	*C.vulgaris*	0.37 ± 0.04 ^b^	0.31 ± 0.04 ^b^	0.28 ± 0.15 ^b^	0.26 ± 0.11 ^b^	0.42 ± 0.01 ^a^	0.29 ± 0.06 ^b^

Value shown are mean of three replicates with ± SD. ^a–c^ Mean values in same row with different superscripts are significantly different (*p* < 0.05) using ANOVA and Turkey post-hoc.

**Table 4 molecules-26-00653-t004:** The division rate, *k* of *N. oceanica, C. vulgaris* on control, 105 °C, 105 °C twice, 121 °C, 121 °C twice and 24 hours’ soil extraction (SE) from *Raja Musa* Forest Reserve (RM) and *Ayer Hitam* Forest Reserve (AH).

Types of SE	Microalgae	Control			Modified SE		
			Media + 105 °C	Media + 105 °C Twice	Media + 121 °C	Media + 121 °C Twice	Media + 24 h
**RM SE**	*N. oceanica*	0.17 ± 0.00 ^a^	0.03 ± 0.09 ^a^	0.03 ± 0.08 ^a^	0.06 ± 0.11 ^a^	0.04 ± 0.05 ^a^	0.10 ± 0.13 ^a^
	*C. vulgaris*	0.12 ± 0.02 ^a^	0.13 ± 0.07 ^a^	0.01 ± 0.23 ^a^	0.19 ± 0.05 ^a^	0.19 ± 0.23 ^a^	0.21 ± 0.07 ^a^
**AH SE**	*N. oceanica*	0.02 ± 0.05 ^a^	0.14 ± 0.05 ^a^	0.12 ± 0.06 ^a^	0.13 ± 0.05 ^a^	0.12 ± 0.06 ^a^	0.12 ± 0.09 ^a^
	*C. vulgaris*	0.12 ± 0.02 ^a^	0.13 ± 0.14 ^a^	0.01 ± 0.15 ^a^	0.18 ± 0.13 ^a^	0.19 ± 0.36 ^a^	0.21 ± 0.09 ^a^

Values shown are mean of three replicates with ^±^ SD. ^a^ Mean value in same row with similar superscripts are significantly different (*p* < 0.05) using ANOVA and Turkey post-hoc.

**Table 5 molecules-26-00653-t005:** Soil extraction methods used in this study.

Extraction Methods	Procedure
Natural Extraction 1 h (NE-1 h)	Soil extracted at room temperature for 1 h
Natural Extraction 4 h (NE-4 h)	Soil extracted at room temperature for 4 h
Natural Extraction 24 h (NE-24 h)	Soil extracted at room temperature for 24 h
Autoclave 105 °C (105 °C)	Soil extracted at 105 °C autoclave for 1 h
Autoclave 121 °C (121 °C)	Soil extracted at 121 °C autoclave for 1 h
Autoclave 105 °C after Natural Extraction 24 h (NE + 105 °C)	Soil extracted at 105 °C autoclave for 1 h before soil extracted at room temperature for 24 h
Autoclave 121 °C after Natural Extraction 24 h (NE + 121 °C)	Soil extracted at 121 °C autoclave for 1 h before soil extracted at room temperature for 24 h
Autoclave 105 °C twice (×2) (105 °C × 2)	Soil extracted at 105 °C autoclave for 1 h, and after cooled (~30 min), soil extracted at 105 °C autoclave for 1 h
Autoclave 121 °C twice (×2) (121 °C × 2)	Soil extracted at 121 °C autoclave for 1 h, and after cooled (~30 min), soil extracted at 121 °C autoclave for 1 h

## Data Availability

Not applicable.

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
