# Peer review of "Kinetics Growth and Recovery of Valuable Nutrients from Selangor Peat Swamp and Pristine Forest Soils Using Different Extraction Methods as Potential Microalgae Growth Enhancers"

_molecules, 2021, doi:10.3390/molecules26030653_

Round 1

Reviewer 1 Report

Manuscript Number: molecules-1075625

Comments and Suggestions for Authors

This study demonstrated the effects of TDN, TDP, and DOC from virgin soils in enhancing microalgal growth. This manuscript also evaluated the changes in soil nutrient concentrations due to various extraction methods. The use of nutrients from virgin soils for microalgal biomass feedstock is an interesting topic. Although the authors provided a detailed discussion, still some points need attention to improve the study. Checking overall english expressions is recomended. There are nominal descriptions that make it difficult to judge whether the content of the discussion is logically described and thus should be revised substantially. Overall the manuscript is acceptable for publication in Molecules, subject to the improvement of the following aspects. I suggest major revision before its consideration for publication.

  1. It is recommended to follow the author's instructions of the journal
  2. In the abstract provide the results with actual quantitative values of how much increase or reduction occurred with the applied
  3. Avoid using abbreviations in abstract and at the start of sentences (line 30,31), also check other places too.
  4. The content delivered in the introduction from “There are mainly two methods…. for the incubation of microalgae” is unnecessary and does not set the background for the said study. Instead of extraction methods, we suggest authors add about how these nutrients can play role in enhancing the microalgal productivity.
  5. There should be a gap statement between lines 69-71 describing the novelty and need for the study.
  6. Authors should better develop a hypothesis, how this work is different from other works in the field.
  7. Can the authors suggest which extraction method performs best in having high nutrient recovery, in abstract and conclusion part as well? This would be a good outcome of this study.
  8. Rephrase line (99-100).
  9. The authors nicely presented the data in figures, however, figure quality can be improved. Adjust figure axis and shape outlines. Figure 1, 2,3, and 8.
  10. Figure and table legends should provide standalone information. Authors should carefully check and improved legends throughout the MS.
  11. The parameters this study selected are rigorous and various, some part of the data could be moved to supplementary information, if possible.
  12. The objective of the study was to evaluate the effects of TDN, TDP, and DOC from soil extracts on microalgae growth, why KNO3 was also added to the Conway media? It creates ambiguity if the growth was enhanced due to nutrients from soil extract.
  13. Where the ammonia came from? Is it from soil extract or from NH4Cl from Convoy media?
  14. Why the OD at 0 day in Fig. 6 and Fig. 7 are different. Will not it affect the growth comparison? Moreover, the X-scale in Fig. 7a is different. Please use the same scale bar.
  15. The reason to present the ratios of TDN, TDP, and DOC as RMFR/AHFR are unclear when the blend of RMFR/AHFR was not subjected to microalgal growth effects.
  16. It would be good if authors present the TDN, TDP and DOC concentrations from two sources in the same figures as in Fig. 8. It would make the comparison easy for readers.
  17. The results section is enough to describe the details. Authors should check what kind of data should be reported in the main text and what should be moved to supplementary materials.
  18. The discussion section is provided with authors own perceptions and are not supported by suitable references. I suggest authors to add recent references to support their findings, meanwhile, the discussion od mechanisms should be explored deeper.
  19. Authors should delete the general assumptions in the conclusion section and report only the findings of the present study.

Reviewer 2 Report

This is a interesting manuscript illustrate the effects of nutrients from Selangor Peat Swamp and Pristine Forest soils using different extraction methods as potential microalgae growth enhancers. Authors may consider following comments and response accordingly.

  1. The abstract of a good journal paper always ends outlining the benefits of the study findings and recommendations as a way forward. The manuscript is missing such 1-2 lines in the abstract.
  2. Graphical abstract would be very useful for the reader. It would help in understanding the authors' intentions and the scheme of the research work carried out.
  3. Keyword numbering is unnecessary. They should be removed.
  4. Introduction. There is no need for paragraph spacing. They should be removed.
  5. The authors should much more explicitly specify the novelty of their work. What progress against the most recent state-of-the-art similar studies was made in this study? Mention this in the revised manuscript sections, including abstract, introduction, and conclusions. Only one sentence: “This is the first study to the best of our knowledge where reserved forest soils were used as nutrient sources for microalgae growth” it is not enough.
  6. What was the aim of the research has not been clearly stated?
  7. Why were these soil types selected for research?
  8. The introduction should show the reader more what the authors' research brings to the commonly known knowledge, which inspired them to plan and implement them, and what new they bring to science. This is completely missing and needs to be completed.
  9. The authors did not formulate any research hypotheses. This should be the starting point for research planning. What did they expect? What were they trying to verify? Needs to be completed.
  10. Introduction is written very roughly. The reader obtains knowledge from handbooks rather than specific scientific facts that became the basis for planning research work.
  11. Can the authors carry out a recalculation of the opical desity to dry weight using known relationships between these biomass concentration indicators, which will enable comparative analysis with other studies? It can be added in results section.
  12. Author should also pay more attention to the practical implications of this study, outlining the challenges in the current research, future work, and recommendations. It is recommended to discuss and explain the appropriate policies based on the findings of this study. In addition, the results should be further elaborated to show how they could be used for real applications.
  13. Is it feasible on a large scale? Is it technically feasible outside the laboratory? Is it even possible? Does it make sense? Will it be profitable? What about the soil after process, what about the waste? There are many problems to discuss.
  14. Figures 1-3. Please place the bars from a and b in one figure. It will be better for comparison. If not, adjust the scale on the graphs to the displayed values. It is illegible as it stands.
  15. The relationships and correlations between the initial nitrogen, carbon and phosphorus concentration and the obtained biomass concentration should be determined.
  16. Author should also consider interpret data in terms of impact from interaction between two nutrients. Since the experiment design is a full factorial design, author should be able to analysis the data via ANOVA contain both main factors and interactions.
  17. Authors should try to optimize the cultivation conditions. The formulas that can predict the biomass concentration depending on the culture medium characteristic will be indicated. For example multiple regression model using a stepwise progressive regression algorithm can be used to identify the relevant predictor variables in the formulas, among the investigated variables.
  18. Did the authors use experimental planning methods?
  19. Figure 6 and 7. Optical density is below 0. Why?

Reviewer 3 Report

Dear Authors,

thank you for your interesting manuscript. In introduction is missing wide overview about selected nutrients eg. 10.3390/agronomy10101513;10.1016/j.scitotenv.2020.141747; 10.1016/j.niox.2020.04.001; 10.1016/j.scitotenv.2020.142168; 10.22034/gjesm.2021.01.08; 10.3390/molecules25214924; sure this references could be use in Discussion.

Line 57- 59 says: ”…Soil nutrients such as phosphorus and nitrogen as well as trace metals are stored, transformed, and cycled in the soil. These nutrients have been extracted and used as soil extracts to culture many soil bacteria [3-5] as well as microalgae [6-9]...” Do the authors mean with this phrase that there have been other attempts to use nutrients in the soil as medium to grow microalgae? If that is the case, please update the references since at least references7-9 don’t refer to this type of test. If not please clarify.

Starting from line 211 I was able to find repeated phrases, for instance, the following affirmation appears twice (once in line 211 and once in line 220) “…The highest concentration of ammonia and nitrate was detected when extraction method of 121 °C twice is employed which achieved a concentration of 2 mg N/L and 35 mg N/L for ammonia and nitrate respectively…” Please recheck this.

Whenever p values are reported the authors need to clarify if the P-value corresponds to the Tuckey Post hoc, ANOVA, or t-test, at the same time they should report the value of the statistic: t statistic, f value, etc, and the degrees of freedom.

Regarding the P-value, the authors should show the actual value and not only an indication of the P-value being higher or lower than the alfa. (e. g) In lines 251, 253, 255, 257…the authors only mention if it was above or below 0,05 “…The statistical analysis showed significant different at P<0.05 as compared to control medium. The growth pattern of C. vulgaris in all modified AH SE is approximately similar with the control experiments (Figure 6B) with insignificant results (P>0.05). In RM SE, N. ocenica has recorded a rise in OD value up to 0.5 which indicated a higher growth rate in control experiment (Figure 7A), where non-significant results (P.0.05) were observed in this study…

In the Sample Collection and Preparation item, I would suggest reevaluating the maps. Currently, they are not very informative, both of them can be very confusing for people not familiar with the geography of Malaysia.

It would be interesting to know if the authors also have additional physicochemical parameters of the extracts such as pH or salinity that could be also influencing the algal cells.

In the methodology section, I recommend clarifying more the statistical analysis to state clearly to state which data was used in which test, please extent this section to a more detailed description of how the analysis was done.

Round 2

Reviewer 2 Report

Manuscript has been improved according  with my remarks. In my opinion paper can be publish in present form.

Author Response

Thank you for the acceptance.

Reviewer 3 Report

Dear Authors, thank you for your revision.

  1. You used only few information and references to improve Introduction - I again recommend wide overview about selected nutrients - see R1
  2. Again carefully check order of references and duplicity (eg. 12 and 16 - are same in the list).
  3. Fig. 6,7 - some problems with numbers of days; fig 1-3,10 improve quality;
  4. Fig 4,5 - significance is missing (post-hoc tests), asterix?
  5. Fig 8 - improve quality, some letters are out or not readable
